# Efficient and selective *N*-alkylation of amines with alcohols catalysed by manganese pincer complexes

Saravanakumar Elangovan[1,2], Jacob Neumann[1], Jean-Baptiste Sortais[2], Kathrin Junge[1], Christophe Darcel[2] & Matthias Beller[1]

Borrowing hydrogen (or hydrogen autotransfer) reactions represent straightforward and sustainable C–N bond-forming processes. In general, precious metal-based catalysts are employed for this effective transformation. In recent years, the use of earth abundant and cheap non-noble metal catalysts for this process attracted considerable attention in the scientific community. Here we show that the selective *N*-alkylation of amines with alcohols can be catalysed by defined PNP manganese pincer complexes. A variety of substituted anilines are monoalkylated with different (hetero)aromatic and aliphatic alcohols even in the presence of other sensitive reducible functional groups. As a special highlight, we report the chemoselective monomethylation of primary amines using methanol under mild conditions.

[1] Leibniz-Institut für Katalyse e.V. an der Universität Rostock, Albert-Einstein-Straße 29a, Rostock 18059, Germany. [2] UMR 6226 CNRS—Université de Rennes 1 Institut des Sciences Chimiques de Rennes, Team Organometallics: Materials and Catalysis—Centre for Catalysis and Green Chemistry, campus de Beaulieu, 35042 Rennes, France. Correspondence and requests for materials should be addressed to M.B. (email: matthias.beller@catalysis.de).

The atom-efficient creation of carbon–nitrogen bonds represents a key step for the synthesis of a plethora of compounds, which are widely applied in life sciences and the chemical industry[1,2]. Due to the importance of these products, there is a number of powerful methods available that address the synthesis of amines, for example, classic nucleophilic substitutions, as well as Buchwald–Hartwig[3], Ullmann reactions[4] and hydroaminations[5]. Most of these more advanced procedures require catalysts which are based on specific (noble metal) complexes. Although these methods proved their efficiency in numerous examples, they often suffer from the co-production of considerable amounts of side products or waste (Fig. 1a). To overcome this drawback, in recent years so-called borrowing hydrogen or hydrogen autotransfer methodologies have been developed which allow for a more sustainable production of amines from alcohols[6–12].

More specifically, in the latter process an initial dehydrogenation (oxidation) of the alcohol leads to the corresponding aldehyde, which subsequently undergoes reductive amination to yield the desired amine. Advantageously, no additional external hydrogen source is needed in this domino process because the parent alcohol acts as the hydrogen donor (Fig. 2). In addition, it should be noted that a variety of alcohols are easily available from renewable feedstock making this methodology especially suitable for the valorization of biomass or biomass-derived building blocks[13].

Pioneering reports dealing with the N-alkylation of amines by alcohols in the presence of homogeneous catalysts were described independently by Watanabe[14] and Grigg[15] at the beginning of the 1980s. Since 2000, tremendous progress has been made in this field using specific precious metal complexes mainly based on Ru[16–20] and Ir[21–25]. Clearly, in terms of sustainability, the use of more eco-friendly, inexpensive and widely abundant metals for the synthesis of amines continues to be a long-standing goal of chemical research[26]. In this respect, it is interesting that Saito reported the N-alkylation of amines based on the combination of iron salts and amino acids in 2011 (ref. 27). However, this protocol proceeds via electrophilic activation and does not involve a hydrogen borrowing mechanism. Recently, a significant breakthrough was accomplished by Feringa and Barta in the alkylation of amines with alcohols using Knölker's iron complex[28]. This work stimulated further developments employing iron-based catalysts[29–33] in this field and for C–C bond formation[34,35]. Meanwhile, Kempe and co-workers[36] reported the selective alkylation of aromatic amines using cobalt complexes[37]. Despite all these advancements using non-noble metal complexes, the development of more active earth abundant catalysts, which work at lower temperature and show improved substrate scope is still challenging. More specifically, selective alkylation of unsaturated amines and methylations are desired. Meanwhile, heterogeneous catalysts

are known for this latter transformation[38–40] and are applied in industry for the preparation of methylamines from methanol on bulk scale. Unfortunately, most of these materials need drastic reaction conditions (250–500 °C) making them not suitable for the synthesis of more advanced amines, especially life science products such as pharmaceuticals. However, in this field the use of amines is immense, and most of the drugs known to date belong to the class of amines. Selected examples of pharmaceutically important molecules are shown in Fig. 3.

After iron and titanium, manganese is the third most abundant transition metal in Earth's crust. In fact, millions of tons of its ore are used for manufacturing steel. In biology, it has important functions for the development and metabolism of humans. With no doubt, manganese represents a highly attractive element for the design of new catalysts[41]. Until now, manganese complexes were frequently employed in oxidations, but not used for catalytic reductions, and only special hydrosilylation and electrocatalytic reactions are known[42–45]. To the best of our knowledge, the use of homogeneous manganese catalysts for alkylation of amine has not been reported yet[46,47]. Moreover, manganese-based pincer type complexes and their catalytic applications are sparsely discussed in the literature[48–51]. During the development of this project, Milstein and co-workers reported the selective synthesis of imines by dehydrogenative coupling of alcohols and amines[52]. Complementary to that work, herein we describe the first well-defined manganese complex for N-alkylation of amines with a variety of alcohols including methanol through hydrogen autotransfer methodology.

## Results

**Reaction design.** Due to the abundance and positive properties of manganese, recently we started a programme to explore novel catalysis with such complexes[53]. More specifically, we were attracted by the potential of PNP pincer complexes, which are known for their cooperative catalysis under mild conditions[54,55]. Hence, at the start of this study four different manganese pincer complexes were tested with aniline (1 equiv.) and benzyl alcohol (1.2 equiv.). Gratifyingly, in the presence of 2 mol% of **1** or **2** and 1 equiv. of t-BuOK at 80 °C, N-benzylaniline **5a** was obtained in 78% and 56% yield, respectively (Fig. 4).

In contrast to the recent work of Milstein[52], the corresponding imine was detected only in low amounts (<10%). Mn(II)PNP complex **3** and cationic Mn(I)NNN complex **4** were found to be less active for this N-alkylation reaction. Notably, no product formation was observed, when the reaction was conducted without addition of base using complex **1** even at 140 °C (see Supplementary Table 4, entry 1). We explain this observation by the specific activation of these stable pre-catalysts: in the presence of base, deprotonation of the coordinated amine is believed to occur, which could be followed by the decoordination of the

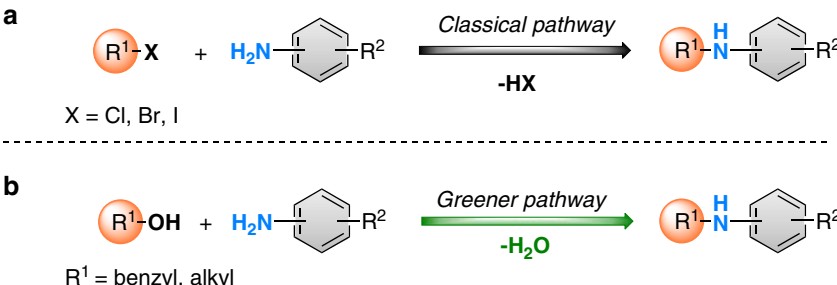

**Figure 1 | Different methods of C–N bond formation.** (**a**) Traditional approach using alkyl halides and (**b**) greener pathway using borrowing hydrogen methodology.

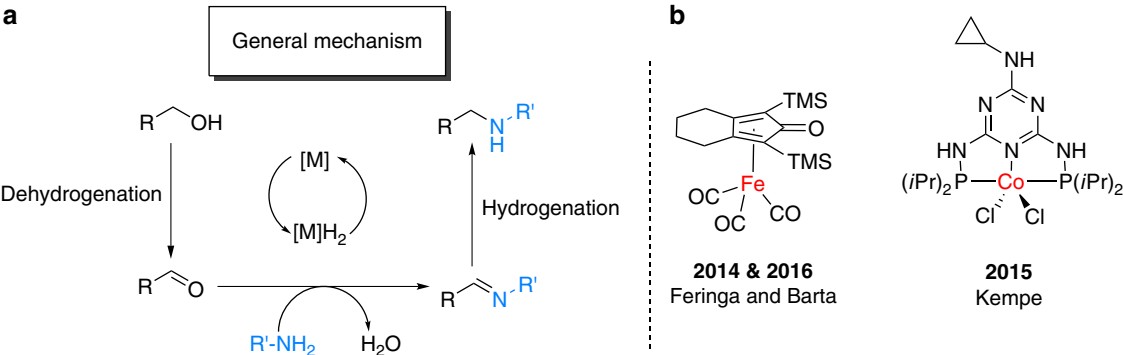

**Figure 2 | N-alkylation of amines.** (**a**) General mechanism and (**b**) recent examples of non-noble metal complexes catalysing the N-alkylation of amines.

**a**

Fluoxetine

Rosiglitazone

Methapyrilene

**b**

Cinacalcet

Resveratrol derivatives

**Figure 3 | Selected examples for nitrogen containing pharmaceuticals.** (**a**) Examples with methylated amine functionalities and (**b**) examples with alkylated amine functionalities.

**1**
91% (78%)

**2**
60% (56%)

**3**
13% (11%)

**4**
12% (7%)

**Figure 4 | N-alkylation of aniline with benzyl alcohol: optimization with different Mn complexes.** Reaction conditions: aniline (0.5 mmol), benzyl alcohol (0.6 mmol), [Mn] (0.01 mmol), t-BuOK (1 equiv.) and toluene (1 ml), 80 °C. Conversion and yield were determined by GC analysis using hexadecane as an internal standard.

halide anion. The resulting amido manganese complex should react with the incoming alcohol to form the corresponding alkoxy manganese species. This is believed to lead to the carbonyl compound via β-hydrogen elimination. Finally, the manganese hydride complex is likely to mediate imine reduction.

Further evaluation of different reaction parameters showed optimal results when performing the reaction in toluene as solvent, with t-BuOK (0.75 equiv.) as the base and a loading of 3 mol% of the complex **1** (see Supplementary Tables 1–4).

Interestingly, under these reaction conditions, the alkylation proceeds highly selective at low temperature (80 °C). Notably, in no case N,N-dialkylation was observed. In the absence of manganese catalyst, no desired product was observed (see Supplementary Table 4, entry 2).

**N-alkylation of various anilines with benzyl alcohols.** For any new catalyst development in this area, it is most important to

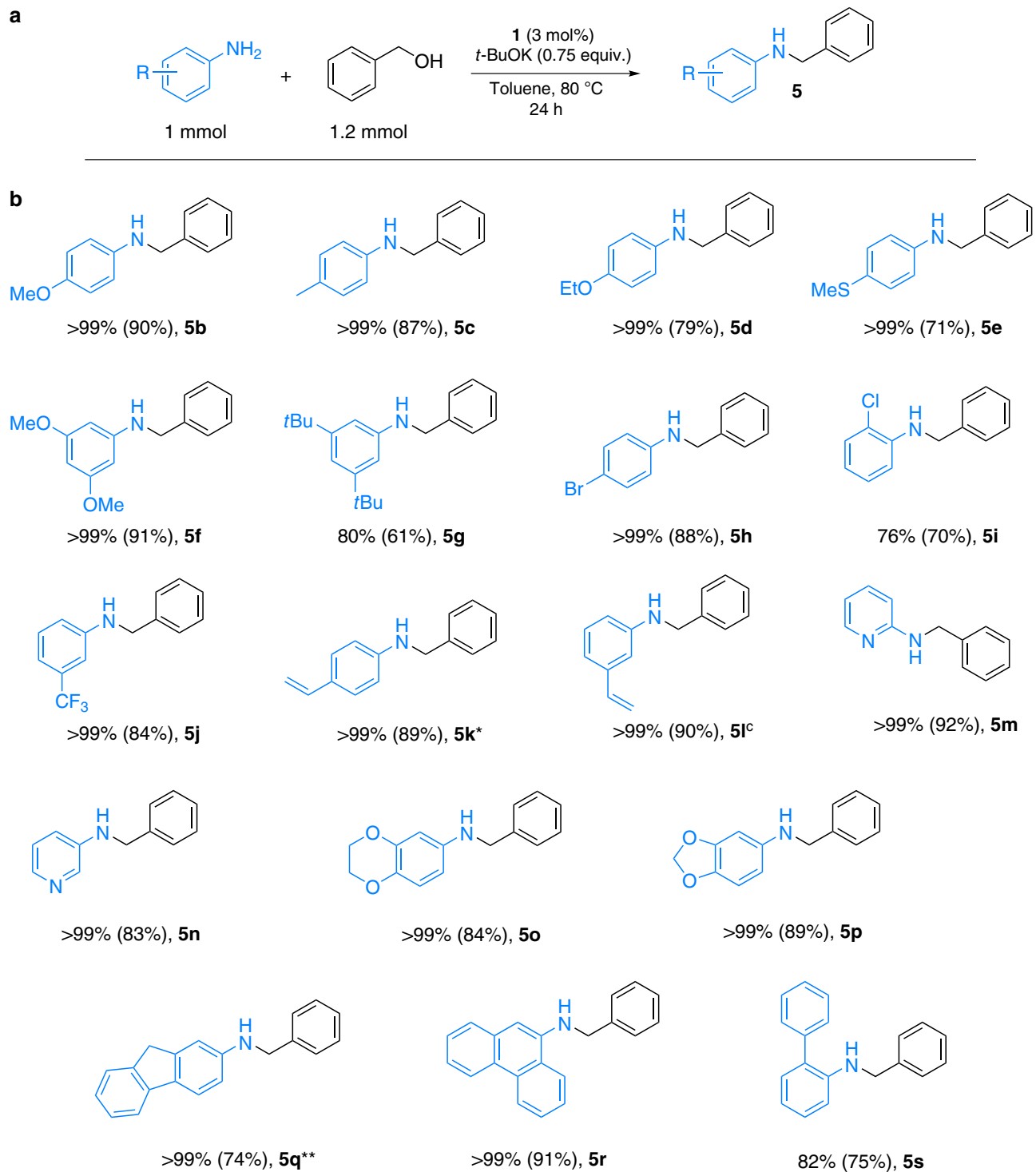

**Figure 5 | Selective *N*-alkylation of various aromatic amines with benzyl alcohol. (a)** General reaction conditions: aniline derivative (1 mmol), benzyl alcohol (1.2 mmol), **1** (3 mol%), *t*-BuOK (0.75 equiv.) and toluene (2 ml), 80 °C, 24 h. **(b)** Reaction of different aniline derivatives with alcohols. Conversion was determined by GC (isolated yield in parentheses). *Traces of reduction (<2%) of double bond were observed. **11% of *N*, 9-dibenzyl-9H-fluoren-2-amine was detected.

show its broad applicability. To demonstrate the usefulness of our novel catalytic system, various aromatic amines and alcohols were tested under optimized conditions. First, we explored the alkylation of different substituted anilines with benzyl alcohol. Substrates bearing both electron-donating (Fig. 5, **5b–d**, **5g**) and electron-withdrawing substituents on the aryl ring of aniline were selectively alkylated to afford the *N*-monoalkylated anilines in good yields (typically 80–90%). Even sterically hindered, *ortho*-chloro aniline (**5i**) and di-meta-*tert*-butyl aniline (**5g**) provided the corresponding monoalkylated amines with excellent selectivity. Advantageously to noble metal catalysts and the reported cobalt and iron complexes[28,35], amines containing C=C bonds were alkylated smoothly to **5k–5l** with high chemoselectivity. Notably, heteroaromatic amines led to the compounds **5m**

and **5n** in good isolated yields (92% and 83%, respectively). Furthermore, important building blocks for pharmaceuticals, for example, aminobenzodioxane derivatives, are effectively transformed (**5o–5p**). To demonstrate the synthetic utility of this method, the alkylation of 2-aminopyridine with benzyl alcohol was also performed on gram scale and led to **5m** in 89% isolated yield as a white solid.

**N-alkylation using (hetero)aromatic and aliphatic alcohols.** Next, we explored the possibility to apply different primary alcohols as the coupling partner. To our delight, the reaction of (hetero)aromatic alcohols proceeded smoothly and furnished the desired products in most cases with moderate to good yields (up to 96%, Fig. 6, **6a–6n**). Gratifyingly, biomass-derived furfuryl alcohol was selectively monoalkylated, too (Fig. 6, **6f–6g**). Long (C$_8$) as well as short-chain (C$_2$)-containing aliphatic alcohols were successfully applied to alkylate 2-aminopyridine yielding the corresponding N-alkylated derivatives in 49–93% after 48 h (**6o–6s**).

Resveratrol-derived amines (see Fig. 3) are known to be active for the treatment of Alzheimer's disease[56]. In light of the high chemoselectivity in the presence of vinyl groups, our herein-reported catalytic system seemed to be especially useful. To proof this assumption, 4-aminostilbene was alkylated with different alcohols in the presence of **1** under the standard conditions. Noticeably, in all cases the alkylated derivatives **7a–7e** were isolated in good to excellent yields (88–97%, Fig. 7).

Apart from intermolecular reactions, an intramolecular cyclisation was also performed. Indeed, under the optimized conditions (100 °C, 48 h), 2-(2-aminophenyl)ethanol led to the corresponding indole in 98% yield (Fig. 8).

**Manganese catalysed N-methylation of aniline with methanol.** Among the different aliphatic alcohols, the selective monomethylation of amines with methanol is most challenging. Apart from the formation of side products such as N,N-dimethyl amines, methanol is more difficult to dehydrogenate. However,

**Figure 6 | N-alkylation of (hetero)aromatic amines using (hetero)aromatic and aliphatic alcohols.** (**a**) General reaction conditions: aniline derivative (1 mmol), benzyl alcohol (1.2 mmol), **1** (3 mol%), t-BuOK (0.75 equiv.) and toluene (2 ml), 80 °C, 24 h. (**b**) Conversion was determined by GC (isolated yield in parentheses). **6o–6s** 48 h. *22% of the corresponding imine was observed. **2 equiv. of ethanol was used.

**R = H, 88%, 7a**
**R = Cl, 96%, 7b**

**97%, 7c**

**90%, 7d**

**93%, 7e**

**Figure 7 | Synthesis of resveratrol derivatives.** Reaction conditions: 4-aminostilbene (1 mmol), alcohol (1.2 mmol), **1** (0.03 mmol), *t*-BuOK (0.75 equiv.) and toluene (2 ml), 80 °C, 24 h.

**1** (3 mol%)
*t*-BuOK (1 equiv.)
100 °C, 48 h
Toluene

**98%, 8**

**Figure 8 | Alternative synthesis route for indole.** Alternative synthesis of indole via an intramolecular reaction of 2-aminophenethyl alcohol.

*N*-methylated amines are very important drug molecules and natural products[1]. Therefore, this transformation is still performed using toxic and/or hazardous methyl halides or sulfates in the agent. Despite this importance, to date only limited studies on transition metal-catalysed *N*-methylation of amines using methanol have been described[57–59], mainly employing precious metals such as Ru[60,61] and Ir[62,63]. To the best of our knowledge, there exist no reports on non-noble metal-catalysed *N*-methylation of amines by methanol. Based on our continuing interest of *N*-methylation reactions[64,65], we performed the synthesis of *N*-methylaniline derivatives in the presence of 3 mol% PNP manganese complex **1**. As shown in Fig. 9, a series of anilines gave the *N*-methylated products in good to excellent isolated yields (**9a–9k**: 52–94%). In most cases, the catalyst showed very good selectivity vide supra and even Br- and I-substitutents were well tolerated (**9f, 9i**), albeit in the case of sterically hindered 2-iodo *N*-methyl aniline **9j** some dehalogenation was observed.

Again, heteroaromatic and vinyl-substituted anilines were selectively monomethylated to the corresponding secondary *N*-methylaniline derivatives. As mentioned before, in all the cases we did not observe any traces of dialkylation products. In agreement with these observations, the reaction of *N*-methylaniline with benzyl alcohol did not lead to the tertiary aniline *N*-benzyl-*N*-methylaniline.

## Discussion

In summary, we have demonstrated for the first time that molecular-defined manganese pincer complexes **1–2** (Mn-PNP) are efficient catalysts for the benign inter- and intramolecular formation of C–N bonds. Advantageously, these complexes are highly stable and can be conveniently handled in the presence of air. This novel protocol for *N*-alkylation of aromatic amines with primary alcohols proceeds under mild conditions

(80–100 °C) with excellent chemoselectivity. Hence, this catalytic system tolerates a large variety of functional groups, including olefins, halides, thioether, benzodioxane and heteroaromatic groups. Of special importance are the *N*-methylations of various amines using methanol, which constitute the first examples of this transformation using non-noble metal complexes under mild conditions.

## Methods

**General analytical methods.** Unless otherwise stated, all reactions were performed under an argon atmosphere with exclusion of moisture from reagents and glassware using standard techniques for manipulating air-sensitive compounds. All isolated compounds were characterized by [1]H NMR and [13]C NMR spectroscopy and high resolution mass spectrometry. Nuclear magnetic resonance spectra were recorded on Bruker AV 300 or 400 (Supplementary Figs 1–64). All chemical shifts ($\delta$) are reported in p.p.m. and coupling constants ($J$) in Hz. All chemical shifts are related to residual solvent peaks [CDCl$_3$: 7.26 ([1]H), 77.16 ([13]C); C$_6$D$_6$: 7.16 ([1]H), 128.06 ([13]C); DMSO-d$_6$: 2.5 ([1]H), 39.52 ([13]C)]. All measurements were carried out at room temperature (RT) unless otherwise stated. Mass spectra were in general recorded on a Finnigan MAT 95-XP (Thermo Electron) or on a 6210 Time-of-Flight Liquid Chromatography/Mass Spectrometry (Agilent). Gas chromatography (GC) was performed on a HP 6890 with a HP5 column (Agilent).

**Reagents.** Unless otherwise stated, commercial reagents were used without purification. *t*-BuOK (Aldrich, 99.99%).

**N-alkylation of primary anilines with primary alcohols.** An oven-dried 25-mL Schlenk tube, prepared with a stirring bar, was charged with the yellow complex **1** (14.9 mg, 0.03 mmol), *t*-BuOK (84 mg, 0.75 mmol) and dry toluene (2 ml). Then, the corresponding alcohol (1.2 mmol) and amine (1 mmol) were added to the red coloured suspension. Solid materials were weighed into the Schlenk tube under air, and the Schlenk tube was subsequently connected to a Schlenk line and vacuum-argon exchange was done for three times. Liquid compounds and solvent were charged under an argon flow. The Schlenk tube was placed into an aluminium block and heated to 80 °C and stirred for a given time. The reaction mixture was cooled to RT, quenched with water and extracted with ethyl acetate. The organic phase were then dried over MgSO$_4$ and concentrated under reduced pressure. The residue was purified by flash chromatography on silica gel (*n*-pentane/diethylether) to afford the desired product.

**N-methylation of primary anilines using methanol.** An oven-dried pressure tube was charged with complex **1** (14.9 mg, 0.03 mmol) and *t*-BuOK (112 mg, 1 mmol). Solid amines (1 mmol) were weighed into the pressure tube under air, and the pressure tube was connected to a Schlenk line and vacuum-argon exchange was performed three times. Liquid amines (1 mmol) and dry, degassed methanol (1 ml) were charged under an argon stream after the three vacuum-argon exchanges. The pressure tube was closed with a stopper and was heated to 100 °C. After 24 h, the reaction mixture was cooled to RT and extracted with ethyl acetate/water. The organic layer was dried over MgSO$_4$ and transferred into a round bottom flask. SiO$_2$ (350 mg) was added to the mixture. The solvent was removed *in vacuo* and

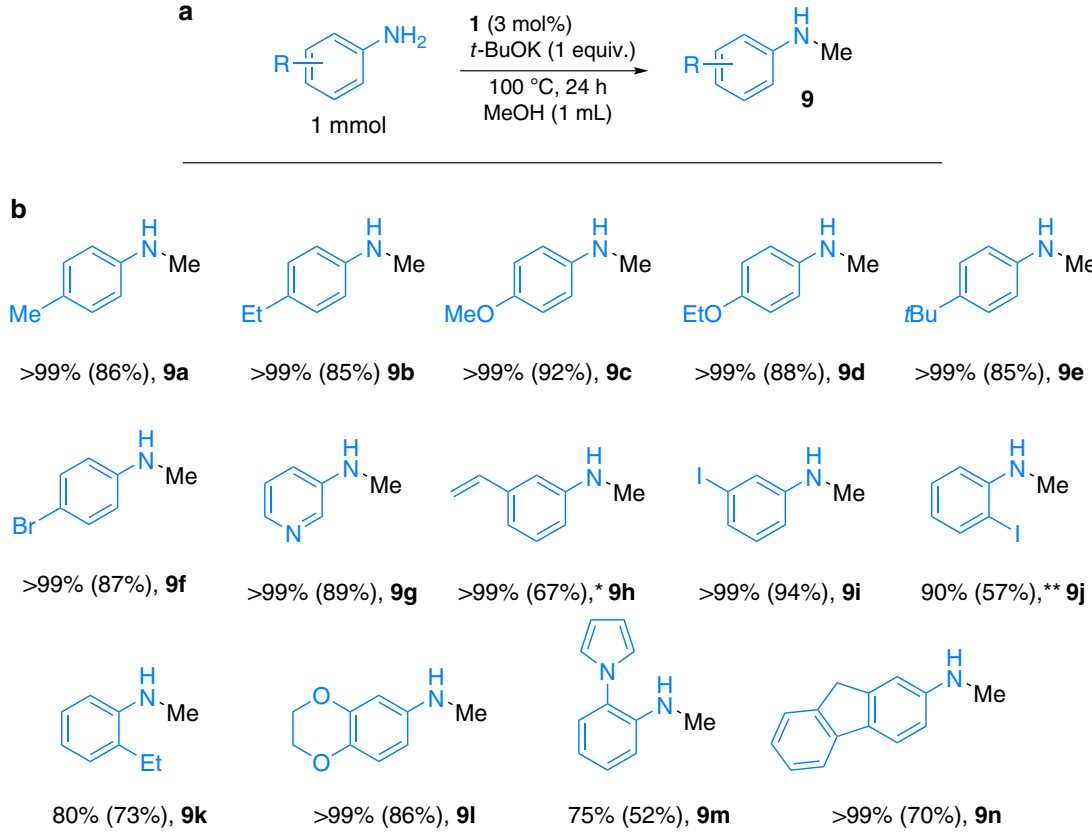

**Figure 9 | *N*-methylation of primary anilines using methanol. (a)** General reaction conditions: aniline derivative (1 mmol), **1** (3 mol%), *t*-BuOK (1 equiv.), and toluene (2 ml), 100 °C, 24 h. **(b)** Conversion was determined by GC ( isolated yield in parentheses). *Traces of reduction of double bond. **15% dehalogenation was observed.

the product was purified by column chromatography using heptane and ethyl acetate.

**Data availability.** All data are available from the authors upon reasonable request.

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

## Acknowledgements

We thank Dr C. Fischer, S. Buchholz and S. Schareina (all at the LIKAT) for their excellent technical and analytical support. C.D. and J.B.S. thank the CNRS and the University of Rennes 1 for financial support. S.E thanks the 'Region Bretagne' for the PhD fellowship.

## Author contributions

S.E. and M.B. planned the project. S.E. and J.N. performed the experiments and analysed the results. M.B., C.D, J.B.S. and K.J. guided the research. M.B., S.E., K.J., J.N., C.D. and J.B.S. participated in writing the manuscript.

## Additional information

**Competing financial interests:** The authors declare no competing financial interests.

