## [Peer review file · Nature Communications]

Reviewers' comments:

Reviewer #1 (Remarks to the Author):

This manuscript by Beller and co-workers describes a three-step alcohol to alkylamine pathway which proceeds via dehydrogenation, Schiff-base condensation with aniline, and hydrogenation of the resulting imine using a PNP pincer-supported Mn precatalyst. The novelty of this work is somewhat diminished due to a recent article by Milstein and co-workers which describes an Mn-catalyzed alcohol dehydrogenation/Schiff-base condensation (JACS, 2016, 138, 4298-4301). However, it should be noted that the efforts described herein are superior in many ways to the reactivity reported by Milstein. For example, the (PNP)Mn precatalyst in this work is active at 80-100 {degree sign}C (vs. 135 {degree sign}C) and it is somewhat remarkable that in situ imine hydrogenation is observed under such low pressures of dihydrogen (Mn mediated hydrogenation is far less explored than Fe/Co/Ni hydrogenation). I believe the catalytic claims made in this article will be viewed as the benchmark to surpass for base-metal catalyzed amine alkylation, and together with the Milstein article, this work is likely to inspire others in the field to screen catalysts for analogous reactivity.

It should be noted that the quality of the experimental work as it relates to characterization of the organic products is excellent. However, there are clear and obvious deficiencies in the interpretation of catalyst activation and the characterization of precatalysts 1-4. The conclusions, references, and quality of manuscript composition are all sufficient for publication. Ultimately, the results do warrant publication in Nature Communications; however, serious attention must be paid to the following points prior to the article being accepted:

- On page 3, the authors incorrectly state that Mn is the second most abundant element in the earth's crust. Quick inspection reveals that it isn't in the top ten (O, Si, Al, Fe, Ca, Na, K, Mg, Ti, H).
- There is no supporting evidence for the information given in Figure 5. To claim that tBuOK deprotonates the amine, the authors need to perform a scaled-up stoichiometric reaction between complex 1 and tBuOK. Given that the dicarbonyl precatalyst described by Milstein possesses an anionic PNP pincer ligand, it is likely that 1a can be isolated if formed. This would be an important observation because salt metathesis between tBuOK and 1 could just as easily yield (PNP)Mn(CO)₂(OtBu), which could promote alcohol dehydrogenation following alkoxide sigma-bond metathesis (i.e., deprotonation of the amine may never occur).
- To further prove that amine deprotonation does occur (as indicated), the authors also need to conduct the stoichiometric dehydrogenation of a deuterated alcohol (methanol-d₄ would be a reasonable/inexpensive choice given the reactivity observed, but it is fairly easy to label any of the substrates used). Recovery of 1c post reaction would reveal amine deuteration (if deprotonation occurs) by ¹H/²H NMR spectroscopy.
- It is odd that near stoichiometric KOtBu is required for the reaction to reach completion, when 3 mol% should be sufficient. Is the excess KOtBu serving to dry the substrates, in turn preventing decomposition?
- Attention must be paid to inorganic characterization before this article can be published. The authors propose complexes 1b and 1c but do nothing to characterize or spectroscopically identify them. Unless this is done, it might be better to remove Figure 5 and its discussion until a true mechanistic investigation can be conducted. The spectroscopic characterization of complex 2 has several question marks instead of values for J-coupling. No NMR data is given for complex 3 and complex 4 is completely ignored in the SI. Supporting IR/NMR data is not shown for all compounds and elemental analysis data has not been provided.

Overall, my recommendation would be to publish this article following moderate-to-significant revision.

Reviewer #2 (Remarks to the Author):

The authors report on Mn complex catalyzed alkylation of amines by alcohols. The alkylation of amines by alcohols is a sustainable transformation, which has received a lot of attention in recent years. Most of the catalysts are based on Ir and Ru and the authors contributed to this chemistry significantly. As mentioned by the authors, base metal catalysts have been introduced already. Here, the first example of a Mn catalyst permitting this reaction is described. A PNP pincer Mn complex was used. In addition, the scope is impressive, most interesting to me, the selective methylation (applying methanol as the alkylating agent). The manuscript will most likely be a door opener to Mn complex catalysis and I feel that the contribution should be published in a high-impact journal.

I have a few suggestions that could improve the manuscript.

- 1) Figure 1. b) is green and selective in comparison to a) if R1 is an alkyl and it would be good to define R1 anyway. For aryl substituents, things are different. The phenol based chemistry is nearly not developed for path b.
- 2) I am not sure that Mn is the second most abundant transition metal in the earth crust. To the best of my knowledge, it is titanium.
- 3) For some of the synthesized examples, I see a huge difference in GC and isolated yield, 5e, 5q, 6e, 6m and n to mention just a few. The authors may want to optimize the work up procedures or give reasons why this huge difference is unavoidable.
- 4) A more complete citation of the homogeneous Fe and Co catalysts, which catalyze the N alkylation of amines by alcohols efficiently, seems appropriate.

Reviewer #3 (Remarks to the Author):

This paper by Beller and coworkers describes the highly efficient and selective N-alkylation of amines with alcohols using manganese pincer complexes. These results represent the first example of this reaction using manganese complexes, which are attractive replacements for noble metals (such as Ru or Ir) in this (and other) catalytic transformation. Given the highly challenging nature of this reaction and the general lack of methods utilizing abundant, sustainable metals (such as manganese) - the results described here are very important advances in the field.

The use of manganese complexes in reductions (or hydride transfer processes) has not yet been shown, and there is very limited knowledge - therefore the results described here are significant and will open the way to many new applications using similar manganese complexes.

The specific advantages of the methodology are that it is: selective for mono-alkylation of amines with alcohols (an advantage compared to many noble metal systems), it takes place at relatively mild reaction conditions, and a good substrate scope was demonstrated. Notably, this includes the mono-methylation of aromatic amines with methanol, for the first time with a non-noble metal catalyst.

The methodology is valid, the approach is reliable and the experimental data are of high quality.

The references used are appropriate.

The results are presented in a clear manner appropriate for both a specialized and a more general readership.

I recommend the paper to be accepted after the following minor points have been addressed:

-A rather high loading of base has to be used in these reactions. Also there is a large variation in product yield depending on the type of base used. (in supporting info: Table S1, S3) It would be beneficial to reduce the amount of base added. Can the authors comment on possible future

improvements related to this point?

- It would be good for the reader to know the outcome of a blank reaction just with base added, but no Mn-complex (Table S4 entry 2) - a short mention of this could be added to the discussion on page 4/5.

Author responses to reviewer comments

Reviewer 1:

1. On page 3, the authors incorrectly state that Mn is the second most abundant element in the earth's crust. Quick inspection reveals that it isn't in the top ten (O, Si, Al, Fe, Ca, Na, K, Mg, Ti, H).

The referee is right. We apologize for this mistake. Hence, we have changed the sentence in the manuscript as follows: "After iron and titanium, manganese is the third most abundant transition metal in Earth's crust."

2. There is no supporting evidence for the information given in Figure 5. To claim that tBuOK deprotonates the amine, the authors need to perform a scaled-up stoichiometric reaction between complex **1** and tBuOK. Given that the dicarbonyl precatalyst described by Milstein possesses an anionic PNP pincer ligand, it is likely that **1a** can be isolated if formed. This would be an important observation because salt metathesis between t-BuOK and **1** could just as easily yield (PNP)Mn(CO)₂(OtBu), which could promote alcohol dehydrogenation following alkoxide sigma-bond metathesis (i.e., deprotonation of the amine may never occur).

We thank the referee for this suggestion. As the amine is coordinated to Mn, the pKa is significantly lower than the corresponding free amine and t-BuOK is strong enough to deprotonate it. According to the reviewer suggestion, we conducted the stoichiometric reaction [Condition: **1** (0.06 mmol, t-BuOK (0.18 mmol), benzene d₆, RT, 1 h) between complex **1** and t-BuOK yielding the amido complex **1a**. This also agrees well with DFT calculations.

In addition, we carried out a reaction with a derivative of complex **1**, which has a methyl group at the nitrogen instead of a proton. Using this complex, under the optimized reaction conditions (0.03 mmol (PNMeP)Mn(CO)₂Br, t-BuOK (0.75 mmol), aniline (1 mmol), benzyl alcohol (1.2 mmol), toluene, 80 °C, 24 h) we only observed 5% *N*-benzylideneaniline, which is the same result that we obtained using complex **1** at 140°C without the addition of base (Table S4, entry 1). This observation clearly indicates that the N-H is important for the formation of the catalytic active species.

3. To further prove that amine deprotonation does occur (as indicated), the authors also need to conduct the stoichiometric dehydrogenation of a deuterated alcohol (methanol-d₄ would be a reasonable/inexpensive choice given the reactivity observed, but it is fairly easy to label any of the substrates used). Recovery of **1c** post reaction would reveal amine deuteration (if deprotonation occurs) by ¹H/²H NMR spectroscopy.

We thank the referee for his suggestions. Accordingly, we performed a number of experiments including the stoichiometric dehydrogenation of methanol-d₄. Unfortunately, we were not able to isolate or to spectroscopically characterize the expected complex **1c**.

4. It is odd that near stoichiometric KOtBu is required for the reaction to reach completion, when 3 mol% should be sufficient. Is the excess KOtBu serving to dry the substrates, in turn preventing decomposition?

A minimum of 0.5-0.75 equiv. of base are necessary to obtain a reasonably efficient reaction within 24 h. We don't think that the base acts mainly as a drying agent. Indeed, we have performed the benchmark reaction in the presence of 0,2 equiv of *t*-BuOK with and without molecular sieves. No significant differences in conversion and yield were observed these results were included in the supporting information (Table S4, entry 16). In addition, we investigated whether it is possible to bring the reaction to complete conversion in the presence of 20 mol% of KOtBu at longer reaction time. Indeed, the catalyst is still active after 72 h (yield at that time: 47%). These results were included in the supporting information (Table S4, Entry 15).

Apparently, even if a catalytic amount of base is necessary to activate the pre-catalyst by NH deprotonation of the ligand to generate the active species, the excess of base is beneficial for a more efficient catalytic cycle.

5. Attention must be paid to inorganic characterization before this article can be published. The authors propose complexes **1b** and **1c** but do nothing to characterize or spectroscopically identify them. Unless this is done, it might be better to remove Figure 5 and its discussion until a true mechanistic investigation can be conducted.

We thank the referee for his comments. As stated correctly it would be beneficial to unambiguously characterize the expected complexes **1b** and **1c**. Unfortunately, as stated above we were not able to characterize those two complexes. Therefore, as suggested by the referee we removed Figure 5 from our manuscript.

6. The spectroscopic characterization of complex 2 has several question marks instead of values for J-coupling. No NMR data is given for complex 3 and complex 4 is completely ignored in the SI. Supporting IR/NMR data is not shown for all compounds and elemental analysis data has not been provided.

Again, we thank the referee for his suggestion. As requested, we have completed the characterization of complexes **3** and **4**. We have included the analytical data for complex **4** in the supporting information. We also included the HR-MS and elemental analysis data in the supporting information for the complex **3**. Due to the paramagnetic nature of the complex **3**, it was impossible to obtain useful NMR data.

Reviewer 2

1. Figure 1. b) is green and selective in comparison to a) if R1 is an alkyl and it would be good to define R1 anyway. For aryl substituents, things are different. The phenol based chemistry is nearly not developed for path b.

We thank the referee for his suggestion and have corrected this and included in the manuscript.

2. I am not sure that Mn is the second most abundant transition metal in the earth crust. To the best of my knowledge, it is titanium.

We apologize for this mistake. The referee is right. Now we have modified the sentence in the manuscript. (see referee 1)

3. For some of the synthesized examples, I see a huge different in GC and isolated yield, 5e, 5q, 6e, 6m and 6n to mention just a few. The authors may want to optimize the work up procedures or give reasons why this huge difference is unavoidable.

For the products 5e, 6m and 6n, due to tedious separation (mainly from the starting unreacted alcohols and the obtained amines), the yields of purified amines is lower than GC conversion.

For the fluorene derivative 5q, 11% of N, 9-dibenzyl-9H-fluorene-2-amine was observed as a side-product. We included this observation in the footnote of Figure 6.

For the reaction leading to 6e, 22% of the corresponding imine was observed in the crude reaction mixture. We included this remark in the footnote of the Figure 7.

4. A more complete citation of the homogeneous Fe and Co catalysts, which catalyze the *N*-alkylation of amines by alcohols efficiently, seems appropriate.

We thank the referee for this suggestion. We have added more data and cited the following references in the manuscript.

1. Bala, M., Verma, P. K., Sharma, U., Kumar, N. & Singh, B. Iron phthalocyanine as an efficient and versatile catalyst for *N*-alkylation of heterocyclic amines with alcohols: one-pot synthesis of 2-substituted benzimidazoles, benzothiazoles and benzoxazoles. *Green Chem.* **15**, 1687-1693 (2013).

2. Zhang, G., Yin, Z. & Zheng, S. Cobalt catalyzed *N*-alkylation of amines with alcohols. *Org. Lett.* **18**, 300-303 (2016).

Reviewer 3:

1. -A rather high loading of base has to be used in these reactions. Also there is a large variation in product yield depending on the type of base used. (in supporting info: Table S1, S3) It would be beneficial to reduce the amount of base added. Can the authors comment on possible future improvements related to this point?

*See comment 4 for referee 2. Furthermore, depending of the basicity, there are significant differences in activity, the strongest used bases giving the best yields (*t*-BuOK, 78% and *t*-BuONa, 45%).*

2. It would be good for the reader to know the outcome of a blank reaction just with base added, but no Mn-complex (Table S4 entry 2) - a short mention of this could be added to the discussion on page 4/5.

Thank you for this suggestion. We have included the following sentence in the manuscript: "In the absence of manganese catalyst, no desired product was observed (Table S4, entry 2)".

REVIEWERS' COMMENTS:

Reviewer #1 (Remarks to the Author):

Upon revision, Beller and co-workers have eliminated several inaccuracies and have done a nice job of filling in the previously requested information. After attention is paid to the following minor points, this article should be suitable for publication.

- The authors reference Figure 1a when mentioning side products/waste, although none are shown. Could Figure 1a better show the benefits of the greener pathway?
- Line 39 should read "alcohols are available from renewable feedstocks". Minor grammatical errors like this exist throughout the manuscript and are likely to be caught upon further editing/review.
- In line 84, the authors may want to remove "complexes" or replace "catalyzed" with "for".
- The authors decided to remove the mechanism rather than include the supporting information they have gathered (regarding activation to form 1a). If lines 109-114 are going to be left in the paper, the authors should approach these statements with care. For example, "base addition is believed to result in deprotonation and concurrent salt formation to generate the respective amido complex. This complex is believed to react with incoming alcohol to generate the alkoxide, which undergoes B-H elimination to yield the aldehyde. After condensation occurs, the resulting Mn-H complex is likely to mediate imine hydrogenation." As is, the authors are stating all of this as fact, without presenting suitable mechanistic evidence or referencing precedent.
- In line 214, "seconday" is spelled incorrectly.